# Molecular basis of human CD22 function and therapeutic targeting

June Ereño-Orbea[1], Taylor Sicard[1,2], Hong Cui[1], Mohammad T. Mazhab-Jafari[1], Samir Benlekbir[1], Alba Guarné[3], John L. Rubinstein[1,2,4] & Jean-Philippe Julien[1,2,5]

CD22 maintains a baseline level of B-cell inhibition to keep humoral immunity in check. As a B-cell-restricted antigen, CD22 is targeted in therapies against dysregulated B cells that cause autoimmune diseases and blood cancers. Here we report the crystal structure of human CD22 at 2.1 Å resolution, which reveals that specificity for α2-6 sialic acid ligands is dictated by a pre-formed β-hairpin as a unique mode of recognition across sialic acid-binding immunoglobulin-type lectins. The CD22 ectodomain adopts an extended conformation that facilitates concomitant CD22 nanocluster formation on B cells and binding to trans ligands to avert autoimmunity in mammals. We structurally delineate the CD22 site targeted by the therapeutic antibody epratuzumab at 3.1 Å resolution and determine a critical role for CD22 N-linked glycosylation in antibody engagement. Our studies provide molecular insights into mechanisms governing B-cell inhibition and valuable clues for the design of immune modulators in B-cell dysfunction.

---

[1] Program in Molecular Medicine, The Hospital for Sick Children Research Institute, Toronto, ON, Canada M5G 0A4. [2] Department of Biochemistry, University of Toronto, Toronto, ON, Canada M5S 1A8. [3] Department of Biochemistry and Biomedical Science, McMaster University, Hamilton, ON, Canada L8S 4L8. [4] Department of Medical Biophysics, University of Toronto, Toronto, ON, Canada M5G 1L7. [5] Department of Immunology, University of Toronto, Toronto, ON, Canada M5S 1A8. June Ereño-Orbea and Taylor Sicard contributed equally to this work. Correspondence and requests for materials should be addressed to J.-P.J. (email: jean-philippe.julien@sickkids.ca)

Sialic acid-binding immunoglobulin-like lectin (Siglec) receptors are a family of 14 cell surface transmembrane proteins that bind specifically to sialic acid (Sia)-containing glycans, facilitating cell adhesion and/or cell signaling[1]. Siglecs are found primarily in vertebrates on a wide range of immune cells including granulocytes, neutrophils, monocytes, dendritic cells, eosinophils, mast cells, T cells, and B cells[2]. Their functions are determined by their cellular distribution and ligand specificity. One of the best described Siglecs is CD22 (Siglec-2), whose expression is restricted to B cells[3]. CD22 plays a critical role in establishing a baseline level of B-cell inhibition, and thus is a critical determinant of homeostasis in humoral immunity. As a result, CD22 knockout mice have an increased incidence of autoimmune disease and hyperactive B cells[4].

CD22 is a single-spanning membrane glycoprotein of 140 kDa on the surface of B cells. The extracellular domain (ECD) of CD22 is comprised of seven immunoglobulin (Ig) domains (d1–d7) and 12 putative N-linked glycosylation sites. The most N-terminal domain (d1) is of predicted V-type Ig-like fold and recognizes Sias containing $\alpha$2,6-linkages[5]. While human CD22 binds preferentially to Sia N-acetyl neuraminic acid (Neu5Ac), murine CD22 has higher specificity toward the non-human N-glycolyl neuraminic acid (Neu5Gc)[6], highlighting species-dependent specificities for CD22 ligand recognition. Moreover, cell surface sialylated glycans can be modified,

typically at the 4, 6, 7, or 9 hydroxyl positions, which can alter their binding specificities to CD22[7, 8]. Some of these changes are associated with cellular dysregulation. As examples, O-acetylation at the 9 hydroxyl position has been implicated in autoimmunity[7, 8] and in progression of childhood acute lymphoblastic leukemia[9].

CD22, itself sialylated, forms homo-oligomers in *cis* on the surface of B cells[10]. CD22 oligomers are located in dynamic nanoclusters and create a signal threshold of antigen binding that must be achieved prior to B-cell activation[11]. CD22 activity is mediated through the intracellular recruitment of phosphatases that facilitate dephosphorylation of stimulatory co-receptors[12]. CD45 has also been implicated as a CD22 ligand in *cis*[11, 13, 14]. In addition, *trans* engagement of Sia-containing ligands on antigen-bearing cells results in the recruitment of CD22 to the immunological synapse and inhibits BCR signaling in response to self-antigens[15].

The inhibitory function of CD22 and its restricted expression on B cells makes CD22 an attractive target for B-cell depletion in cases of autoimmune diseases and B-cell-derived malignancies. Numerous therapeutic approaches in development harness B-cell inhibition through CD22 to induce tolerance or anergy[16], or to deplete dysregulated B cells through CD22 targeting by either small molecules[17, 18] or antibody–drug conjugates[19]. Constitutive CD22 clathrin-mediated endocytosis[20] allows for the targeted

**Table 1 Crystallographic data collection and refinement statistics**

| | CD22$_{20-330, 5A}$ native (5VKJ) | CD22$_{20-330, 5A}$ HgCl$_2$ | | | CD22$_{20-330, 5A}$ + $\alpha$2-6 sialyllactose (5VKM) | Epratuzumab Fab (5VKK) | CD22$_{20-330, 4Q}$ + epratuzumab Fab (5VL3) |
|---|---|---|---|---|---|---|---|
| *Data collection* | | | | | | | |
| Space group | C2 | C2 | | | C2 | P1 | P1 |
| Cell dimensions | | | | | | | |
| *a, b, c* (Å) | 126.8, 56.6, 49.4 | 126.8, 56.6, 49.1 | | | 124.3, 57.9, 48.1 | 56.7, 61.5, 65.3 | 87.1, 90.2, 136.8 |
| $\alpha, \beta, \gamma$ (°) | 90, 110.7, 90 | 90, 110.2, 90 | | | 90, 107.0, 90 | 71.8, 81.1, 75.9 | 109.4, 93.2, 99.0 |
| | | Peak | Inflection | Remote | | | |
| Wavelength | 0.97949 | 1.0051 | 1.0083 | 1.0033 | 0.97949 | 0.97949 | 0.97949 |
| Resolution (Å)[a] | 46.24–2.12 (2.20–2.12) | 46.24–2.30 (2.40–2.30) | 46.24–2.30 (2.40–2.30) | 46.24–2.40 (2.50–2.40) | 32.70–2.20 (2.28–2.20) | 32.94–2.01 (2.10–2.01) | 39.19–3.10 (3.20–3.10) |
| $R_{merge}$ | 0.063 (0.475) | 0.049 (0.464) | 0.044 (0.459) | 0.048 (0.464) | 0.106 (0.615) | 0.094 (0.551) | 0.090 (0.390) |
| $R_{pim}$ | 0.037 (0.282) | 0.022 (0.229) | 0.019 (0.229) | 0.021 (0.214) | 0.041 (0.321) | 0.088 (0.402) | 0.090 (0.390) |
| $I/\sigma(I)$ | 15.8 (2.3) | 15.8 (1.7) | 18.8 (1.8) | 17.4 (2.0) | 9.9 (1.6) | 9.1 (1.6) | 8.2 (1.9) |
| $CC_{1/2}$ | 99.9 (80.1) | 99.9 (83.9) | 99.9 (85.1) | 99.9 (86.1) | 99.5 (59.7) | 98.7 (52.4) | 98.7 (74.0) |
| Completeness (%) | 99.8 (99.8) | 95.8 (74.9) | 95.5 (73.4) | 98.7 (90.1) | 99.6 (99.8) | 97.0 (94.6) | 94.2 (96.7) |
| Redundancy | 3.8 (3.8) | 3.6 (2.1) | 3.6 (2.1) | 3.8 (2.8) | 3.8 (3.8) | 2.6 (2.8) | 1.9 (1.9) |
| *Refinement* | | | | | | | |
| Resolution (Å) | 46–2.1 | | | | 33–2.2 | 33–2.0 | 39–3.1 |
| No. reflections | 18,725 (1848) | | | | 16,679 (1681) | 52,019 (5050) | 65,849 (6779) |
| $R_{work}/R_{free}$ | 0.194/0.223 | | | | 0.217/0.256 | 0.206/0.248 | 0.277/0.298 |
| No. atoms | 2678 | | | | 2604 | 7260 | 22,916 |
| Protein | 2462 | | | | 2436 | 6680 | 22,598 |
| Hetero | 85 | | | | 99 | 12 | 318 |
| Water | 143 | | | | 53 | 514 | 0 |
| *B* factors | | | | | | | |
| Protein | 39.8 | | | | 49.2 | 36.8 | 76.9 |
| Hetero | 54.0 | | | | 63.3 | 42.8 | 102.7 |
| Water | 42.2 | | | | 46.0 | 40.0 | NA |
| r.m.s deviations | | | | | | | |
| Bond lengths (Å) | 0.002 | | | | 0.005 | 0.007 | 0.006 |
| Bond angles (°) | 0.57 | | | | 0.78 | 0.91 | 1.32 |

*NA* not applicable
[a]Values in parentheses are for highest-resolution shell

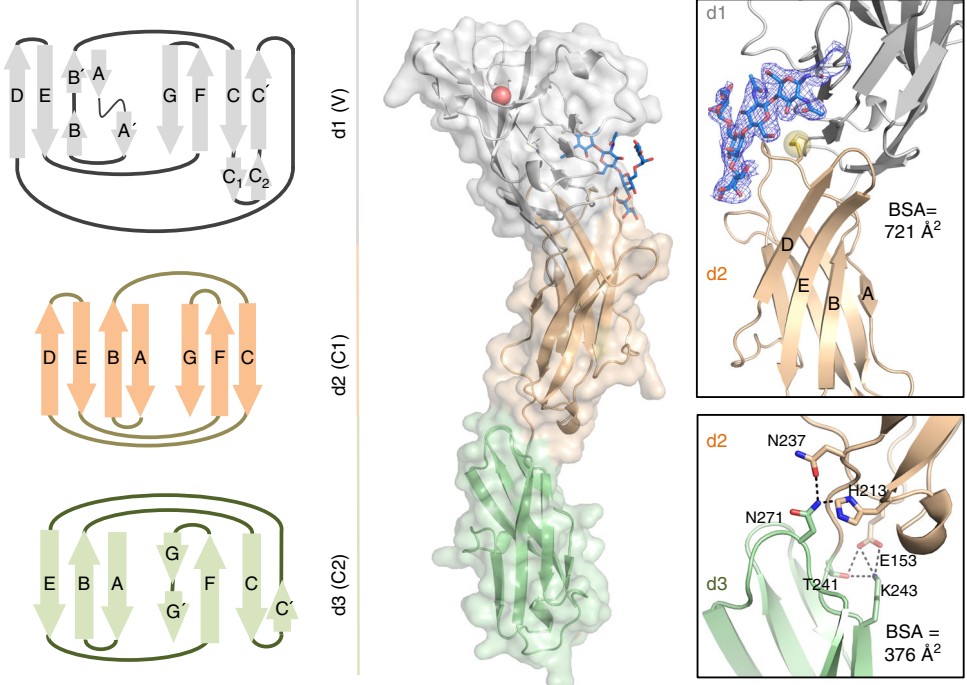

**Fig. 1** Three-dimensional structure of human CD22. Crystal structure of the three N-terminal Ig domains of human CD22 (d1–d3) represented as a schematic diagram, secondary structure cartoon, and transparent surface. d1 (*gray*) adopts a V-type Ig domain fold and R120 known to bind Sia is depicted as a *red sphere*. d2 (*wheat*) adopts an unexpected C1-type Ig domain fold, while d3 (*green*) is of C2-type fold. Insets show inter-domain interfaces of d1/d2 (*top*) and d2/d3 (*bottom*). CD22 d2 strands D–E are unusually elongated and contribute to the extensive d1–d2 interface. A *blue mesh* represents the composite omit electron density map (1.0 σ contour level) associated with the N101 glycan (*sticks*) involved in shaping the d1/d2 interface. Inter-domain disulphide bond C39-C167, highly conserved among Siglec family members, is shown as a *yellow sphere*

delivery of immunotoxins to treat B-cell-related autoimmune diseases and blood cancers[21].

The negative regulation of BCR signaling by CD22 is well understood from mice studies[4, 22] and molecular co-localization imaging[11], but poorly delineated at the atomic level. Using X-ray crystallography, single-particle electron microscopy (EM) and small-angle X-ray scattering (SAXS) techniques, we solved the molecular structure of the extracellular portion of human CD22 alone and in complex with its ligand α2-6 sialyllactose. Our structural analysis of the full-length extracellular portion of CD22 reveals that the CD22 ECD adopts an extended conformation with low flexibility optimally configured to form nanoclusters and interact with self-ligands at the immune synapse. We also structurally delineate the CD22 site targeted by the therapeutic antibody epratuzumab and determine a critical role for CD22 N-linked glycosylation in therapeutic antibody engagement, with potential implications for CD22 recognition on dysfunctional B cells.

## Results

**Crystal structure of the human CD22 ectodomain.** To facilitate crystallization of CD22 ECD, we created a truncated construct that contains the first three Ig domains (residues 20–330) with five of the six N-linked glycosylation sites mutated to alanines (5A: N67A, N112A, N135A, N164A, N231A) (Supplementary Fig. 1). Alanine mutation at N101 disrupted protein expression, pointing to a role in folding for the glycan at this position (Supplementary Fig. 1). Crystals of the CD22$_{20-330,5A}$ construct diffracted X-rays to 2.1 Å resolution and the crystal structure was solved by multiple anomalous dispersion using mercury-soaked crystals (Table 1).

The three N-terminal domains of CD22 are arranged as beads on a string and extend ~110 Å (Fig. 1). As expected, d1 adopts a

V-type fold. Unexpectedly, d2 adopts a C1-type fold, rather than the predicted C2-type fold (Fig. 1). Siglec-4, myelin-associated glycoprotein (MAG), was also reported to have a C1-type fold for d2[23] (Supplementary Fig. 2a). The d2 of Siglec-5 adopts the predicted C2-type fold (Supplementary Fig. 2a)[24]. Thus, the CD22 structure further highlights a heterogeneity in the V-type/d2 structural dispositions among Siglec family members, which might be dictated by whether they are members of classic or CD33-related Siglecs sub-classes.

CD22 has elongated D and E strands in d2 that generate a remarkably large interface area with d1 and extensively stabilize the orientation of the ligand binding domain (Fig. 1). As such, d2 of CD22 buries 721 Å² of surface area on d1, which is substantially greater than that for Siglec-4 (684 Å²) and Siglec-5 (461 Å²) (Supplementary Fig. 2a)[25]. Within the D–E loop of d1, clear electron density is observed for the N-linked glycan at N101, accounting for all monosaccharides in the GlcNAc$_2$Man$_3$ core (Fig. 1; Supplementary Fig. 1f). The N101 glycan is positioned in a hydrophobic environment at the d1/d2 junction (Supplementary Fig. 1f), burying 348 Å² of surface area, which helps explain its importance for CD22 expression. Comparison of the human CD22 sequence with Siglec-2 orthologs and Siglecs-4, -5, and -7 reveals that the N101 glycosylation site is well conserved (Fig. 2a). In the crystal structures of Siglecs-4 (PDB ID: 5LFR) and -7 (PDB ID: 1O7S), the equivalent N101 glycan also buries a significant surface area on the protein (589 and 317 Å², respectively)[23, 25, 26]. As predicted, d3 adopts a C2-type Ig domain topology (Fig. 1). The CD22 d2/d3 interface is smaller (376 Å²) and more hydrophilic than the d1/d2 junction, yet contributes significant inter-Ig domain contacts (including extensive H-bonding networks) that stabilize the CD22 d1–d3 N-terminus in a specific disposition (Fig. 1).

The CD22 V-type d1 domain displays unique features compared to other Siglec family members: (1) the C–C′ strands

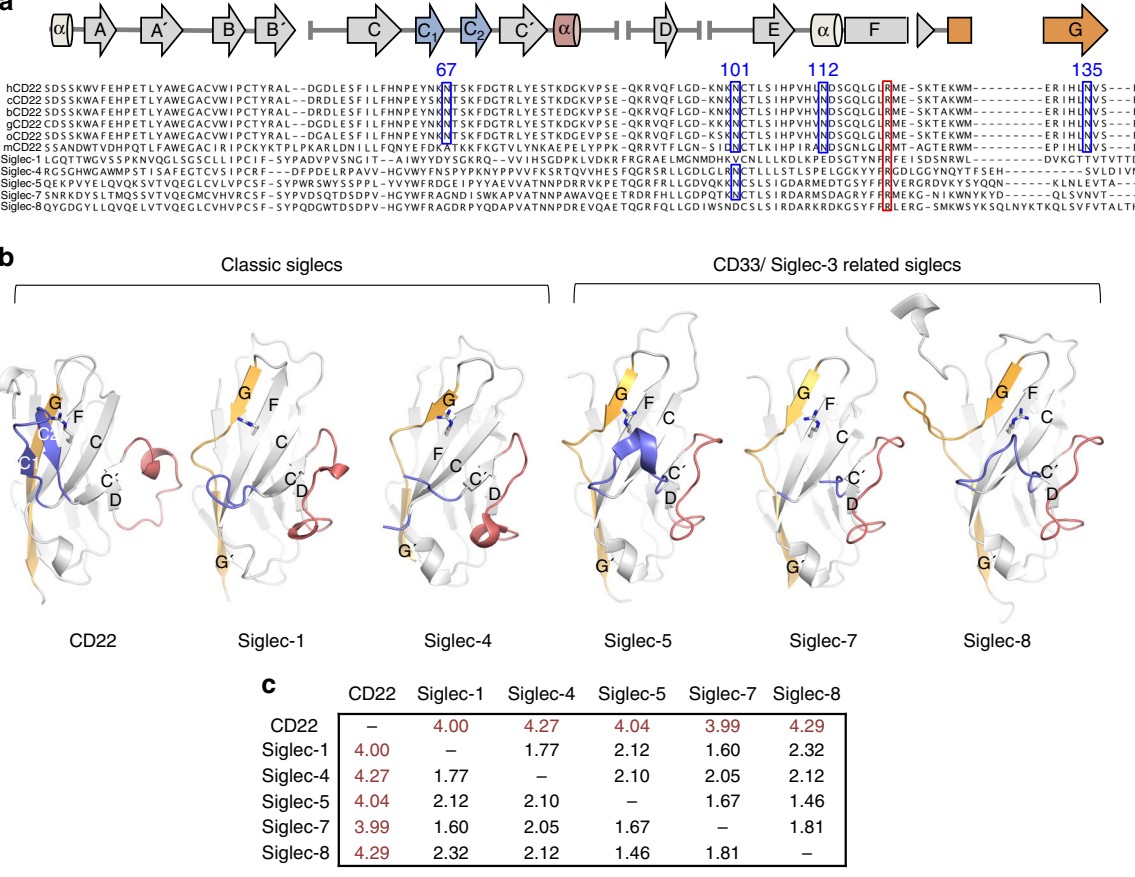

**Fig. 2** Differences in ligand binding domain d1 among the Siglec family. **a** Sequence alignment of the V-type domain of mammalian CD22 and other Siglecs. Siglec -1, -4, -5, -7, and -8. hCD22 secondary structure elements are represented atop the alignment. Putative N-glycosylation sites N67, N101, N112, and N135 are highlighted in *blue boxes*. Conserved R120 involved in binding Sia ligands is marked by a *red box*. **b** Comparison of available three-dimensional structures of V-type domains from classic Siglecs (including the CD22 structure reported here, Siglec-1 (PDB ID: 1QFP)[67] and Siglec-4 (PDB ID: 5LFR)[23]) and CD33/Siglec-3-related Siglecs (Siglec-5 (PDB ID: 2ZG2)[24], Siglec-7 (PDB ID: 1O7S)[26], and Siglec-8 (PDB ID: 2N7A)[51]). Differences in C–C′ (*blue*), C ′–D (*red*), and strand G (*orange*) conformations are highlighted. R120 is shown as *sticks*. **c** Uniqueness of the CD22 V-type fold compared to other Siglecs is evident from elevated Cα r.m.s.d. values calculated with MOE[60]. hCD22 human CD22, cCD22 CD22 from *Pan troglodites* (chimpanzee), bCD22 CD22 from *Pan paniscus* (bonobo), gCD22 CD22 from *Gorilla gorilla* (gorilla), oCD22 CD22 from *Pongo pygmaeous* (orangutan), mCD22 CD22 from *Mus musculus* (mouse)

are elongated and extend into a β-hairpin (hereafter named C1 and C2) that shapes the Sia binding site; (2) strand G is continuous, without a loop insertion; and (3) the C′–D loop protrudes away from the core V-type structure (Fig. 2b). These unique features result in the CD22 V-type domain being the most structurally distant member (highest root-mean-square-deviation (r.m.s.d.)) across Siglecs of known structure (Fig. 2c).

**CD22 specificity for α2-6 sialyllactose ligands**. To understand the structural basis of CD22 specificity for α2-6 Sia ligands, we soaked CD22$_{20–330,5A}$ crystals with α2-6 sialyllactose [Neu5Acα(2-6)Galβ(1-4)Glc] and solved the liganded structure at 2.2 Å resolution (Table 1). The CD22 binding site for α2-6 sialyllactose is formed by strands F and G and loop C–C′ (containing the elongated C1/C2 β-hairpin) (Fig. 3a). No electron density was observed for the glucose moiety of α2-6 sialyllactose in our crystal structure. This is likely due to a lack of interactions for this moiety with CD22. Similar to other Siglecs, the majority of CD22 interactions occur through the Sia portion of the ligand (181 Å$^2$ of buried surface area for Sia out of a total of 276 Å$^2$ for Neu5Acα (2-6)Gal) (Supplementary Table 1). The negatively charged C1 carboxylate of Sia interacts via a salt bridge with the guanidinium

group of the highly conserved R120 (Fig. 3a, b). Substitution of R120 to either A or E completely abrogates binding to α2-6 sialyllactose in isothermal titration calorimetry (ITC), compared to a 281 ± 10 μM binding affinity for WT CD22$_{20–330}$ at 25 °C (Supplementary Fig. 3), as consistent with previous findings[27]. E126 and W128 make key contacts with Sia (Fig. 3a, b), and these interactions corroborate previous biochemical studies that delineated the CD22 binding site by mutagenesis of these residues to lysine and arginine, respectively[28].

The C1/C2 β-hairpin constrains the binding pocket of CD22. Y64 stacks against the CD22 β-hairpin and the hydrophobic face of galactose of α2-6 Sia ligands (Fig. 3c). Y64 thus largely dictates specificity for the α2-6 glycosidic linkage, and not the α2-3 glycosidic linkage (Fig. 3c). The equivalent position of human CD22 Y64 in mouse is F68, making the aromatic ring a conserved feature across CD22 in different species to participate in stacking interactions with α2-6 Sia ligands (Supplementary Fig. 3i). R131 H-bonds with the C2 galactose hydroxyl, further contributing to this specificity. Substitution of R131 to A, K, or Q only marginally impacted the binding affinity to α2-6 sialyllactose (Supplementary Fig. 3c–f), indicating a peripheral role for this residue in mediating CD22 ligand binding. Modeling of an extended (and biologically relevant) Sia-Gal-GlcNAc-Man glycan in our

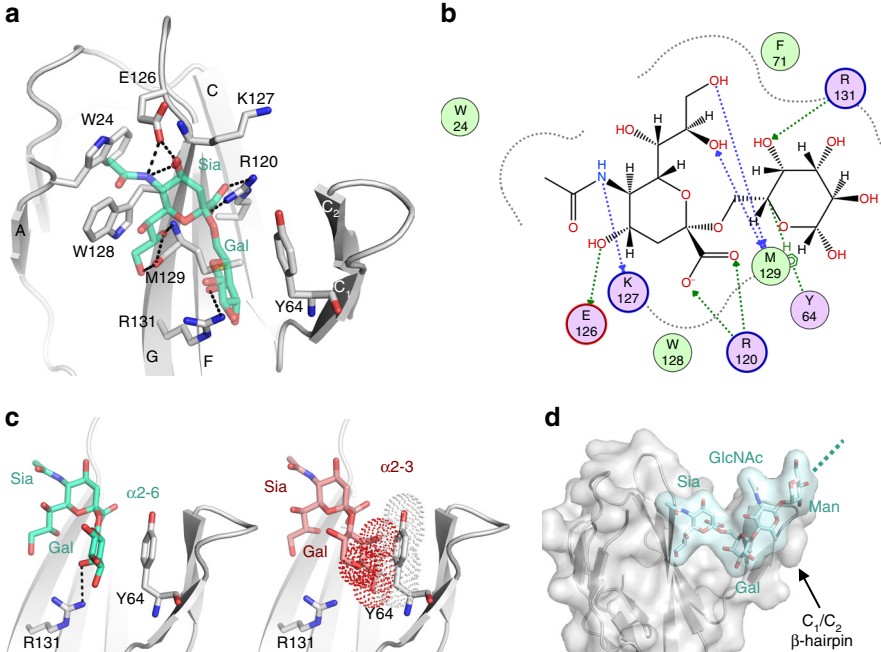

**Fig. 3** CD22 specificity for the α2-6 glycosidic linkage. **a** Sia and Gal moieties (*green*) of the α2-6 sialyllactose ligand extensively H-bond (*black dashed lines*) with human CD22 (*gray*). **b** Schematic representation of the CD22-Neu5Acα(2-6)Gal interaction network. Surface exposed residues are contoured with *gray dashed lines*. *Pink circles* with *black outline*: polar residues, with *red outline*: acidic residues, with *blue outline*: basic residues. *Green circles* represent hydrophobic residues. H-bonds implicating side chains are presented as *green dashed lines*, and implicating main chain atoms in *blue dashed lines*. Rendering generated with MOE[54]. **c** Specificity for the α2-6 glycosidic linkage is largely due to residues Y64 and R131 (*left*). Glycans with α2-3 linkages would clash (overlapping *dotted spheres*) with Y64 (*right*). **d** Modeling of Sia-Gal-GlcNAc-Man (*cyan surface*) reveals that the CD22 $C_1/C_2$ β-hairpin is optimally configured to extensively interact with the branch of a complex N-glycan of α2-6 glycosidic linkage. The *dashed cyan line* indicates the direction of the continued glycan branch

crystal structure reveals that CD22, and particularly the C1/C2 β-hairpin, is optimally configured to extensively interact with branches of complex N-glycans with terminal α2-6 glycosidic linkages (Fig. 3d). We also note that the tip of the C1/C2 β-hairpin itself has a conserved putative N-linked glycan at position N67 (Fig. 2a; Supplementary Fig. 1), which potentially contributes carbohydrate–carbohydrate interactions that further stabilize the CD22 ligand contact[29].

Overall, the complex and unliganded structures of CD22 are highly similar (Cα r.m.s.d. of 0.35 Å for d1), indicating that carbohydrate recognition by CD22 is largely mediated by a preformed binding site (Supplementary Fig. 2b). Extensive intra-molecular H-bonds between C1 and C2 in the β-hairpin and van der Waals interactions between F71 and M129 are major determinants of the preformed binding site (Supplementary Fig. 2c). We note minimal interactions within the crystal lattice that might have artifactually constrained the C1/C2 β-hairpin in our soaking experiments. Conversely, Siglecs-4 and -7 undergo a conformational selection in the C–C′ loop, in which the loop becomes ordered (from a previously unordered state) upon ligand binding (Supplementary Fig. 2b). For CD22, a preformed binding site results in an enthalpy-driven interaction with α2-6 Sia-terminated glycans (Supplementary Fig. 3). Evidently, the C–C′ loop in the Siglec family dictates specificity for terminal moieties of the carbohydrate by adopting one of two different conformations: (i) pointing toward the ligand binding pocket as in CD22, and Siglecs -4, -5, and -8; or (ii) extending away from the ligand binding pocket as in Siglecs -1 and -7 (Supplementary Fig. 2b).

**The CD22 ectodomain adopts a tilted rod-like structure**. The cis/trans binding modes of CD22, together with a multi-Ig domain topology imply that the biological function of CD22

might be facilitated by significant conformational plasticity. Inter-domain flexibility has indeed been observed for other receptors with similar properties, including protein tyrosine phosphatase sigma (RPTPσ) involved in synaptogenesis[30]. We next analyzed the structure of full-length CD22 ECD containing all seven Ig domains (residues 20–687) (Supplementary Fig. 1) by negative-stain EM and SAXS. Images of stained particles (Supplementary Fig. 4a) and corresponding 2D class average images (Fig. 4a) revealed a surprisingly small range of conformations adopted by CD22 molecules, indicating low flexibility. Ab initio 3D models of CD22 ECD were calculated from the EM images[31] (Fig. 4b; Supplementary Fig. 4) and SAXS measurements[32] (Supplementary Figs. 4, 5; Supplementary Table 2). These highly similar low-resolution 3D models obtained from independent techniques confirmed that full-length CD22 ECD behaves as an elongated rod. 2D projections of the ab initio EM reconstruction and the SAXS 3D volume account for nearly all particle images classified in 2D classes (Fig. 4a), further demonstrating a predominant CD22 ECD conformation with limited flexibility. The CD22 d1–d3 domains form an angle of ~120° with respect to d4–d7 domains (Fig. 4b; Supplementary Figs. 4, 5). SAXS experiments of CD22 ECD in the presence of α2-6 sialyllactose showed no apparent conformational change upon ligand binding (Supplementary Fig. 5).

**Molecular recognition of CD22 by therapeutic antibodies**. The localization of CD22 in nanoclusters and its extensive N-linked glycosylation likely impact how CD22 can be targeted therapeutically. Consequently, we next characterized the antigenic surface of CD22 recognized by two leading therapeutic antibodies in clinical trials: epratuzumab and pinatuzumab[19]. Binding competition between epratuzumab and pinatuzumab revealed

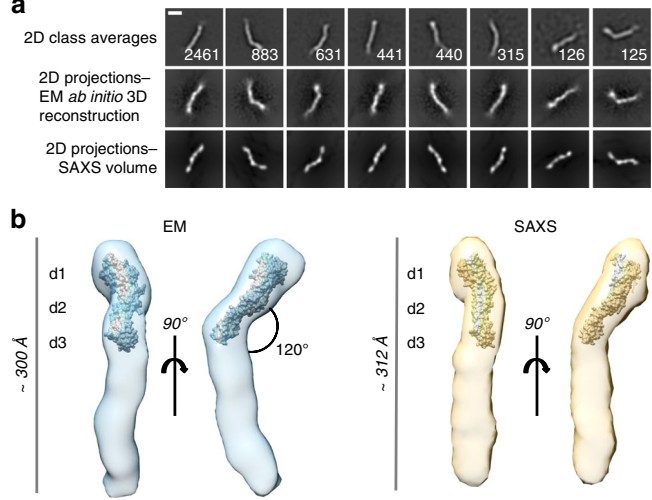

**Fig. 4** The CD22 ectodomain adopts a tilted rod-like structure with low flexibility. **a** The eight primary 2D class average images of CD22 ECD obtained by negative-stain EM (*top*). The number of particle images in each class is indicated. 2D projections of the ab initio EM reconstruction (*middle*) and the SAXS 3D volume (*bottom*) account for nearly all particles classified in 2D classes. *Scale bar* represents 10 nm. **b** Views of the ab initio EM reconstruction (*blue*) and SAXS volume (*yellow*). The crystal structure of CD22$_{20-330,5A}$ is shown as a surface fitted into the EM map and SAXS volume rendered with UCSF Chimera[63]

they recognize non-overlapping epitopes on CD22 (Supplementary Fig. 6a).

To delineate the epratuzumab epitope at high resolution, a glutamine-resurfaced CD22$_{20-330,4Q}$ construct was designed, complexed with epratuzumab Fab, and resulting crystals diffracted to 3.1 Å resolution (Table 1). CD22 d1–d3/epratuzumab Fab crystals were obtained at a relatively low pH (4.6) and we confirmed by biolayer interferometry (BLI) that epratuzumab is capable of binding to CD22 with high affinity in a range of slightly acidic pHs (Supplementary Fig. 6e). The structure corroborated our low-resolution EM data that showed how epratuzumab binds primarily at the base of CD22 d2, with additional interactions with d3 (Fig. 5a; Supplementary Table 3). Our structural findings agree with previous studies[33, 34] that have shown that the epratuzumab epitope is not located in the ligand binding site.

The epratuzumab epitope consists of 1308 Å$^2$ of buried surface area (Supplementary Table 3) and extends beyond only CD22 d3, as previously reported[34, 35]. All three epratuzumab heavy-chain complementarity determining regions (HCDRs), the light-chain CDR1 (LCDR1) and LCDR3 interact with CD22 d2 (Fig. 5b), while HCDR2, LCDR1, and LCDR3 mediate contacts with d3 (Fig. 5c). These interactions were confirmed by mutagenesis in binding experiments (Supplementary Fig. 6b, c). Our 2.0 Å resolution structure of unliganded epratuzumab Fab (Table 1) indicates that its paratope is largely pre-configured for binding its antigenic site (r.m.s.d. of 0.50 Å) (Supplementary Fig. 6d).

Perhaps not surprisingly, the epratuzumab paratope includes an N-linked glycan at position N231, for which clear electron density is observed for the first two GlcNAc residues (Fig. 5d). The CD22 N231Q mutant (a knockout of this glycosylation site) resulted in a 25-fold increase in binding on-rate, and an overall six-fold improvement in binding affinity compared to WT CD22 (Fig. 5e; Supplementary Fig. 7). Consequently, binding kinetics and thermodynamics of epratuzumab Fab to CD22 constructs with different glycan contents revealed an increasing affinity to CD22 with reduced glycan size, with up to a 14-fold improvement

in affinity for smaller glycans (327 vs. 24 nM in BLI; 188 vs. 58 nM in ITC) (Fig. 5e; Supplementary Fig. 7). This effect was not as pronounced for pinatuzumab (Supplementary Fig. 7). A tighter affinity for epratuzumab in the presence of smaller N-linked glycans is primarily due to faster rates of association and a sharp decrease in favorable binding entropy (Supplementary Fig. 7). Together, our data indicate that glycosylation on CD22 impacts the ability of epratuzumab to access its epitope.

## Discussion

CD22 is a B-cell-restricted co-receptor that plays a critical role in the maintenance of B-cell homeostasis. Our structural delineation of CD22 presented here, combined with extensive literature on the function of human and mouse CD22 orthologs, provides an in-depth molecular understanding of its mode of action.

The biology of Sia ligands binding to CD22 is complex; for example, it remains unclear how the availability of Sia ligands in their various glycoforms modulate CD22 function, in both health and disease. Two common animal Sias exist: Neu5Ac and Neu5Gc. They differ by a hydroxyl at the 5′-position that is irreversibly added to Neu5Ac by cytidine monophospho-N-acetyl neuraminic acid hydroxylase (CMAH)[36]. Most mammals, including mouse, mainly express Neu5Gc in their tissues[36]. On the contrary, humans lack the ability to synthesize Neu5Gc because they lack CMAH[37]. Modeling of mouse CD22 interactions with Neu5Gc based on our human CD22 and α2-6 sialyllactose co-crystal structure suggests that a hydroxyl at the 5′-position in Neu5Gc would lead to favorable interactions with E130 (Supplementary Fig. 3h, i). We also note several differences in the composition of the CD22 binding site between human and mouse: P62 and Y64 located in the C1 strand of human CD22 are tyrosine and phenylalanine in mouse CD22, respectively; and R131 located in stand G of human CD22 is a proline in mouse CD22 (Fig. 2a; Supplementary Fig. 3i). These residues likely play a role in the lower affinity of murine CD22 to Neu5Ac[38]. Differences in both circulating ligand glycoforms and binding site chemical composition between human and mouse CD22 highlight the complexities that have evolved for Siglec–ligand interactions across species and the caveats associated with extrapolating findings about CD22 from mouse models to humans.

The predominant ligand recognized by human CD22, Neu5Ac, itself shows structural diversity that arises from N- and O- substitutions, which are of critical importance for ligand recognition and cellular processes[7]. For example, 9-O-acetylation is the most commonly observed Sia substitution and has been linked to autoimmunity in human and mouse models[8]. Modeling suggests that CD22 binding to 9-O-acetylated Sia is sterically impeded by W128 located in strand G of the binding pocket (Supplementary Fig. 3j), providing structural insights into why acetylation of Sia on self-antigens prevents CD22 recognition and increases B-cell-mediated autoimmunity. Future structure/function studies will reveal the spectrum of fine specificities associated with Sia ligand recognition by CD22.

Although the role of CD22 as an inhibitory regulator of BCR signaling is well established, the question of how it binds ligands in both cis and trans is not fully understood. The CD22 preformed binding site and its inter-domain arrangement with low flexibility poses the question of how it binds ligands in both cis and trans. We propose that the binding site in d1 is well positioned to bind flexible N-linked glycans of adjacent CD22 molecules to coordinate clustering of CD22. In this model, the elongated, tilted CD22 structure—and the location of its binding site at the N-terminus—is ideal for inter-molecular interactions with flexible bi-, tri-, and/or tetra-antennary glycans terminated in Sia (Fig. 6).

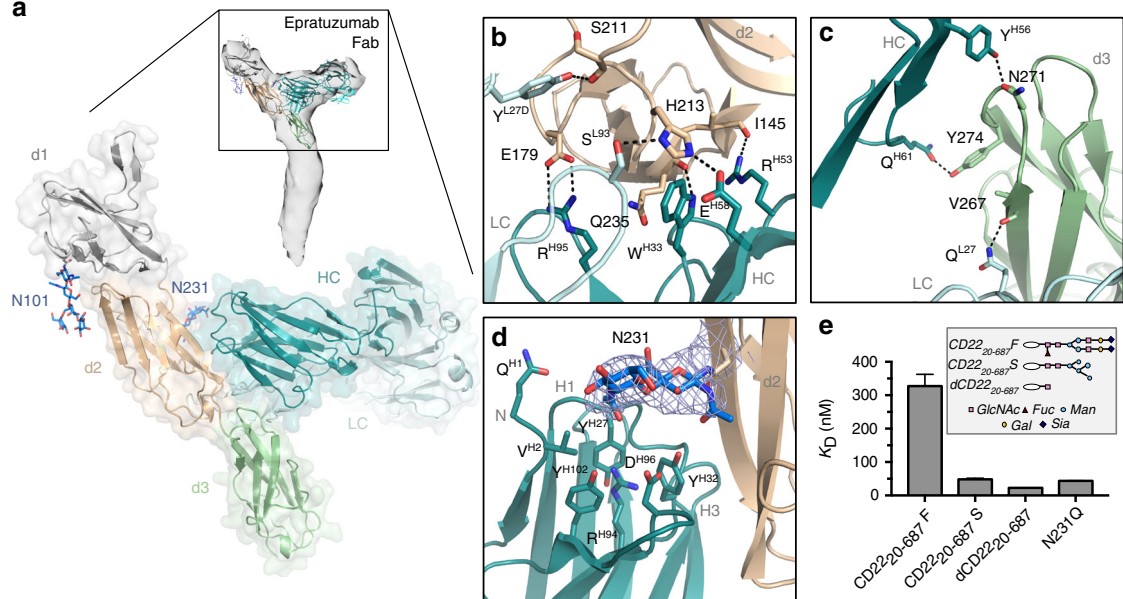

**Fig. 5** Antigenic surface of CD22 recognized by therapeutic antibody epratuzumab. **a** Epratuzumab heavy chain (*dark teal*) and light chain (*light cyan*) bind the CD22 d2 (*wheat*)/d3 (*light green*) interface as revealed by X-ray crystallography (*bottom*) and negative-stain EM (*top*). **b** Epratuzumab interactions with CD22 d2 are mediated by HCDRs, LCDR1, and LCDR3. **c** Epratuzumab interactions with CD22 d3. **d** The N231 glycan is part of the epratuzumab epitope. Composite omit electron density map for the N231 glycan $GlcNAc_2$ residues (*sticks*) is shown as a *blue mesh* (1.0 σ contour level). Epratuzumab residues shown as *sticks* are buried by the N231 glycan. **e** Kinetics of epratuzumab binding to $CD22_{20-687}F$, $CD22_{20-687}S$, $dCD22_{20-687}$, and $CD22_{20-687}F$ N231Q. Epratuzumab binds with higher affinity to CD22 with smaller N-linked glycans. The inset depicts examples of glycoforms likely to be present on CD22 from recombinant expression in HEK293F and HEK293S cells, and treated by EndoH. *Error bars* represent the standard error of the mean (SEM) derived from three independent BLI measurements

Modeling the predicted N-glycosylation sites in our CD22 structure reveals that they are predominantly located on one face of the protein (Fig. 6; Supplementary Fig. 1), facilitating inter-molecular interactions in *cis* and leading to the formation of an interconnected nexus of N-linked glycans atop CD22 nanoclusters (Fig. 6). The ECD of CD22 extends ~300 Å on the surface of B cells, making it optimal in length to also interact selectively with other glycoproteins of similar dimensions, such as protein tyrosine phosphatase CD45 (~220 Å)[39]. Although *cis* ligands occupy CD22 on resting B cells, its relatively weak binding affinity for α2-6 Sia-terminated glycans (~250 μM) would allow its predisposed binding site to dynamically exchange interacting partners in the presence of *trans* ligands on adjacent cells, causing the redistribution of CD22 nanoclusters to the site of cell contact[40]. The length of CD22 may therefore also be optimally evolved to participate in cell–cell recognition through interactions with glycoproteins in *trans*, allowing CD22 recruitment to the immune synapse to surround BCR clusters and sustain B-cell inhibition in the presence of self-antigens[15] (Fig. 6). Our structural insights now enable the structure-based rational design of high-affinity compounds capable of specifically competing with natural ligands to modulate CD22 nanocluster formation, mobility, and co-localization to improve inhibition of over-stimulated B cells[17, 18].

CD22 undergoes clathrin-mediated endocytosis, and is a recycling receptor that can shuttle cargo between the surface of the B cell and the early/sorting endosomes[20, 41, 42]. This mechanism has been exploited for the delivery of antibody conjugated toxins for B-cell depletion[43]. Epratuzumab functions through an alternative mechanism, whereby it acts as a CD22 agonist causing prolonged B-cell inhibition[44], in addition to Fc-dependent receptor trogocytosis[45]. Here, we show that epratuzumab is capable of binding to CD22 with high affinity in a range of pH's, including acidic pH's corresponding to early/sorting endosomes (pH 5.5–6.5) and late endosomes/lysosomes

(pH 4.5–5.5)[46] (Supplementary Fig. 6e). Based on these results, and the very slow off-rates that are observed at the pH of the early/sorting endosomes, it is unlikely that CD22 internalization would lead to dissociation of epratuzumab from CD22, and therefore it is probable that epratuzumab is recycled back to the cell surface alongside CD22, as shown previously for other anti-CD22 antibodies[41]. The recycling of epratuzumab-bound CD22 from the cell surface to the early endosomes may lead to prolonged B-cell inhibition and higher levels of trogocytosis.

Location of the epratuzumab epitope at the d2/d3 interface on a tilted CD22 might ideally promote crosslinking by the IgG, consequently causing accentuated B-cell inhibition and apoptosis as a unique mechanism of action for B-cell depletion through CD22[44] (Supplementary Fig. 7g). Our biophysical and structural data also indicate that glycosylation on CD22 impacts the ability of this therapeutic antibody to access its epitope, while at the same time favorably contributing binding energy to the epratuzumab–CD22 interaction. This hindrance/dependency relationship for binding to heavily glycosylated proteins has previously been described for broadly neutralizing antibodies recognition of the HIV Envelope trimer[47]. A glycan dependency for epratuzumab binding to CD22 is particularly relevant given that variable glycoforms such as truncations or modified branching patterns have been observed on surface glycoproteins in cancer cells due to altered expression of glycosyltranferases[48, 49]. It is currently unclear how the glycosylation patterns of CD22 are altered in B-cell-derived malignancies and autoimmune diseases. Future studies will be required to evaluate whether modifications of the CD22 N231 glycan site on dysfunctional B cells significantly impact on the ability of epratuzumab to engage its protein epitope. Whether the nexus of N-linked glycans atop CD22 nanoclusters offers a yet underappreciated biomarker for dysregulated B cells to be exploited in diagnosis or therapeutic specificities also remains to be determined.

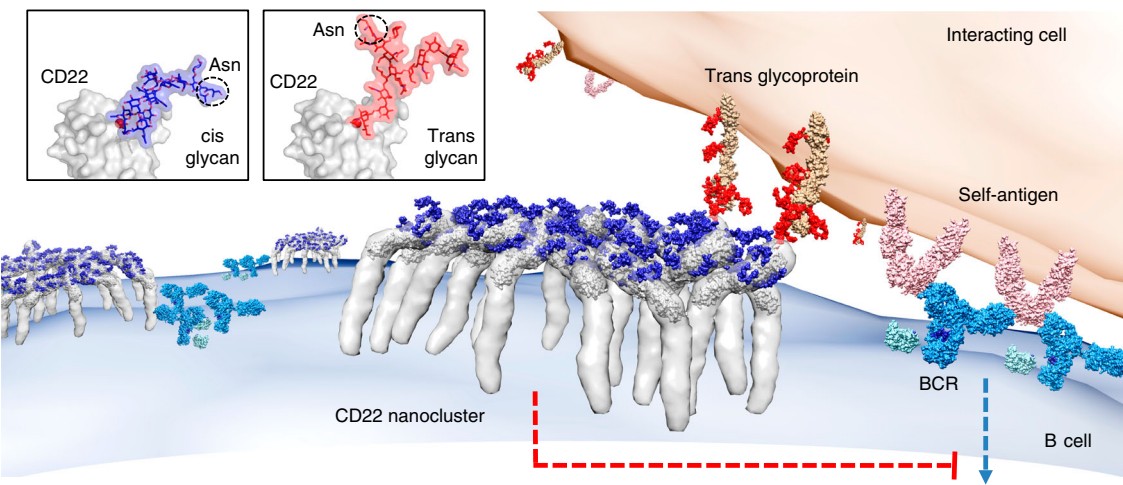

**Fig. 6** Model depicting the ability of the elongated, tilted CD22 structure to form nanoclusters on the surface of B cells. Binding to Sia-terminated glycans in cis (*blue surface*) and trans (*red surface*) is proposed to be mediated by the flexibility of branched complex glycans (*inset*) and the local micro-environment. The CD22 ECD EM envelope is shown as *gray surface*, fitted with the crystal structure of CD22$_{20-330,5A}$. N-linked glycans modeled at all CD22$_{20-330,5A}$-predicted sites are shown as *blue surfaces*. BCRs (*sky blue* and *cyan*) (PDB IDs: 1HZH and 3KH0)[68, 69] are shown binding to self-antigens (*pink*) (PDB ID:4ZXB)[70] on an interacting cell (*salmon surface*). *Blue arrow* represents BCR activation, whereas *red dashed line* represents CD22-mediated BCR inhibition. Engagement of CD22 with trans glycans (*red*) promotes recruitment of CD22 nanoclusters to the immune synapse resulting in B-cell inhibition

## Methods

**CD22 ECD construct design**. Full-length human CD22 ECD (UniprotKB P20273, residues 20–687) was codon-optimized for expression in human cells and synthesized by GeneArt (Life Technologies) (Supplementary Table 4). The construct was subcloned into the pHLsec vector[50] using restriction enzymes *Age*I and *Kpn*I, such that a His$_{6\times}$ tag was at the C terminus of the construct to facilitate affinity purification. Truncated constructs 20–330, 20–504, and 20–592 were PCR amplified and cloned into pHLsec as described for the full-length ECD (Supplementary Table 5).

To test the importance of N-linked glycosylation for CD22 folding, asparagine to alanine point mutants at each of the six predicted glycosylation sites (N67, N101, N112, N135, N164, N231) in the first three domains of CD22 were generated using overlapping PCR (Supplementary Table 5). CD22$_{20-330,5A}$ was generated by mutating asparagine to alanine residues at positions N67, N112, N135, N164, and N231, such that only the glycosylation site at position N101 was retained in the construct. The CD22$_{20-330,4Q}$ construct has N67Q, N112Q, N135Q, and N164Q mutations such that the glycans at N101 and N231 are retained. These constructs were used in crystallization trials alone, and in complex with α2-6 sialyllactose or epratuzumab Fab.

**Expression and purification of CD22**. CD22 constructs were transiently transfected into HEK293F (Thermo Fisher Scientific) or HEK293 Gnt I$^{-/-}$ (HEK293S) (ATCC CRL-3022) suspension cells to produce CD22 glycoforms displaying either glycans with mature carbohydrates (HEK293F), or of high mannose type (HEK293S). Cells were split in 200 ml cultures at $0.8 \times 10^6$ cells per ml. About 50 μg of DNA was filtered and mixed in a 1:1 ratio with transfection reagent FectoPRO (Polyplus Transfections), and incubated at room temperature for 10 min. The DNA:FectoPRO solution was then added directly to the cells, and cells were incubated at 37 °C, 180 rpm, 8% CO$_2$ in a Multitron Pro shaker (Infors HT) for 6–7 days.

Cells were harvested by centrifugation at 6371×*g* for 20 min, and supernatants were retained and filtered using a 0.22 μm Steritop filter (EMD Millipore). Supernatants were passed through a HisTrap Ni-NTA column (GE Healthcare) at 4 ml min$^{-1}$. The column was washed with 20 mM Tris pH 9.0, 150 mM NaCl, 5 mM imidazole buffer prior to elution with an increasing gradient of imidazole (up to 500 mM). Fractions containing CD22 were pooled, concentrated, and separated on a Superdex 200 Increase size exclusion column (GE Healthcare) at 0.5 ml min$^{-1}$ in 20 mM Tris pH 9.0, 150 mM NaCl buffer to achieve size homogeneity. To obtain deglycosylated samples, CD22 expressed in HEK293S cells was treated with the enzyme EndoH (New England Biolabs) for 1 h at 37 °C. Deglycosylated CD22 was purified further via a second size exclusion chromatography.

**Expression and purification of Fabs**. The heavy and light chains of epratuzumab and pinatuzumab Fabs were synthesized by GeneArt (Life Technologies) and subcloned into the pHLsec expression vector[50]. The heavy chain and light chain of each antibody were co-transfected into 200 ml HEK293F cells using FectoPRO (Polyplus Transfections) at a 1:1 ratio of DNA:FectoPRO. Cells were transfected at a cell density of $0.8 \times 10^6$ cells per ml and incubated at 37 °C,

125 rpm, 8% CO$_2$ in a Multitron Pro shaker (Infors HT) for 5–7 days. Cells were harvested and supernatants retained and filtered with a 0.22 μm membrane (EMD Millipore). Supernatants were flowed through a KappaSelect affinity column (GE Healthcare) using an AKTA Start chromatography system (GE Healthcare) and eluted with 100 mM glycine pH 2.2. Eluted fractions were immediately neutralized with 1 M Tris-HCl pH 9.0. Fractions containing protein were pooled and run through a desalting column to change the sample buffer into 20 mM sodium acetate, pH 5.6. Ion exchange chromatography was performed using a MonoS column (GE Healthcare) and eluted with a potassium chloride gradient. Fractions were pooled, concentrated and flowed on a Superdex 200 Increase gel filtration column (GE Healthcare) to obtain purified samples. Peaks were pooled for crystallization trials and binding studies.

**Crystallization and X-ray data collection**. Purified CD22$_{20-330,5A}$ protein was concentrated to 10 mg ml$^{-1}$ in a buffer containing 20 mM Tris pH 9.0 and 150 mM NaCl. Crystals were obtained by sitting drop vapor diffusion in 30% PEG 4000, 0.2 M lithium chloride, and 0.1 M Tris pH 8.5 in 96-well plates after mixing 0.1 and 0.1 μl of protein and solution using an Oryx4 crystallization robot (Douglas Instruments). Crystals were cryo-protected by soaking them in mother liquor solution containing 20% glycerol and flash cooled in liquid nitrogen. X-ray diffraction data were collected at the 08ID and 08BM synchrotron beamlines at the Canadian Light Source (CLS).

Initial attempts to solve the structure of CD22$_{20-330,5A}$ by molecular replacement did not yield any solution using the V- and C2-type Ig domains of homologous Siglec family members as search models[24, 26, 51]. To acquire phasing information, we soaked native crystals with 7 mM of mercuric chloride (Analar) for 30 min. At CLS beamline 08BM, a fluorescence scan was performed near the Hg L-III absorption edge (energy range 11,560–11,570 keV) to determine the appropriate wavelengths for collection of multi-wavelength anomalous dispersion datasets. The three wavelengths selected for multi-wavelength anomalous dispersion were 1.0051 Å at the absorption peak, 1.0083 Å at the inflection point, and 1.0033 Å at remote wavelength. Full multi-wavelength anomalous dispersion datasets were collected on a single crystal (Table 1).

Data for CD22$_{20-330,5A}$ derived with mercury were processed using XDS[52]. Based on the C2 space group, Matthews volume calculation[53] and predicted binding sites of mercury, we estimated one molecule in the asymmetric unit, and expected one anomalous scatterer. Initial phases obtained using AutoSolve[54] were useful for automatic building of the structure by AutoBuild[55]. The heavy atom that allowed phasing by multi-wavelength anomalous dispersion was bound to C308 (an unpaired CD22 cysteine) in d3. Iterative rounds of manual model building in Coot[56] and refinement with Phenix[57] followed, with statistics reported in Table 1. Representative electron density is shown in Supplementary Fig. 8a.

Complex crystals were obtained by soaking native CD22$_{20-330,5A}$ crystals with 25–30 mM α2-6 sialyllactose (Sigma-Aldrich). X-ray diffraction data were collected at CLS at beamline 08ID to 2.2 Å resolution. The crystal structure of the complex CD22$_{20-330,5A}$ with α2-6 sialyllactose was solved by molecular replacement using CD22$_{20-330,5A}$ as a search model in Phaser[58]. Representative electron density is shown in Supplementary Fig. 8b.

To determine the structure of unliganded epratuzumab Fab, purified sample was concentrated and crystals were obtained at 7 mg ml$^{-1}$ by sitting drop diffusion.

Epratuzumab Fab crystallized in a condition containing 85 mM Tris, pH 8.5, 25.5% PEG 4000, 170 mM sodium acetate, and 15% glycerol at 20 °C. Crystals were flash frozen in liquid nitrogen. X-ray diffraction data were collected at the 08ID beamline at CLS and processed in space group P1 using XDS[52]. The structure was solved by molecular replacement in Phaser[58] using a structure from our internal Fab database as a starting model, and was refined by manual building in Coot[56] and using phenix.refine[57].

Purified $CD22_{20-330,4Q}$ was complexed with epratuzumab Fab and purified by size exclusion chromatography in a buffer that contained 20 mM Tris pH 8.0 and 150 mM NaCl. The complex was concentrated to 5 mg ml$^{-1}$ and crystals were obtained by sitting drop vapor diffusion in a condition that contained 80 mM sodium acetate, pH 4.6, 160 mM ammonium sulfate, 20% PEG 4000, 20% glycerol, and 0.01 mg ml$^{-1}$ papain. Crystals were flash frozen in liquid nitrogen, and X-ray diffraction data were collected using synchrotron radiation at the 08ID beamline at CLS. Crystals diffracted to 3.1 Å resolution and the structure was solved by molecular replacement using Phaser[58] with unliganded epratuzumab Fab and $CD22_{20-330,5A}$ structures as search models. The structure was refined by manual building in Coot[56] and using phenix.refine[57]. Representative electron density is shown in Supplementary Fig. 8c.

PyMOL was utilized for structure analysis and figure rendering[59]. All buried surface area values reported were calculated using EMBL-EBI PDBePISA[25]. Molecular operating environment (MOE) was used for molecular modeling and ligand binding interactions[60]. Software was accessed through SBGrid[61].

**Negative-stain electron microscopy.** $CD22_{20-687}$ was stained with 2% uranyl formate. A data set consisting of 100 images was collected manually with a field-emission FEI Tecnai F20 electron microscope operating at 200 kV and an electron exposure of 30 e$^-$ Å$^{-2}$. Images were acquired with an Orius charge-coupled device (CCD) camera (Gatan Inc.) at a calibrated magnification of ×34,483, resulting in a pixel size of 2.61 Å at the specimen and a defocus range of ~0.75–2 µm was used. A total of 6312 particle images were manually selected with Relion 1.3[62]. 2D classification of particle images was performed with 100 classes allowed. Ab initio reconstruction of the molecular envelope of $CD22_{20-687}$ was calculated using cryoSPARC[31]. Five initial 3D classes were generated by stochastic gradient descent from random seeds. Similarly, a negative-stain dataset containing 1879 particle images was collected for $CD22_{20-687}$ in complex with epratuzumab Fab, using parameters as described above. 2D class averages from manually selected particle images were generated in Relion 1.3[62] and 3D envelopes of the structure were determined ab initio with cryoSPARC[31]. 2D projections were calculated using script genproj_evenlyspaced.f90 (https://sites.google.com/site/rubinsteingroup/). Fitting of the CD22 d1–d3 crystal structure in the $CD22_{20-687}$ EM volume using UCSF Chimera[63] favored one orientation (higher fitting score of 0.905 with 0 atoms outside of map contour) compared to when it was fitted in an alternative orientation at the base (0.890 fitting score with 27 atoms outside of map contour). Fitting of the $CD22_{20-330,4Q}$–epratuzumab Fab crystal structure in the $CD22_{20-687}$–epratuzumab Fab EM volume using UCSF Chimera[63] was unambiguous.

**SAXS data collection and processing.** Samples of purified $CD22_{20-687}$ were concentrated to 1.25, 2.5, or 5.0 mg ml$^{-1}$ for studies by SAXS at the Argonne Advanced Photon Source BIO-SAXS beamlines 12-ID-D and 18-ID-D. For each concentration, in the absence and presence of 10 times molar excess of α2-6 sialyllactose, 15 exposures of 2 s each were collected and the scattering curves were generated by subtracting the contribution from the buffer. Analysis of the scattering curves using PRIMUS[64] showed no signs of aggregation, long-range interactions, or radiation damage, but the 1.25 mg ml$^{-1}$ concentration had weak signal (as seen in the Kratky plot [I(q)q2 vs. q], Supplementary Fig. 5b). Agreement between radius of gyration (Rg) and I(0) values determined from the Guinier plot and the pair distribution function (P(r)) further confirmed the good quality of the data. As such, determination of the $D_{max}$, overall shape, excluded volume, and molar mass were performed with high confidence via analysis of the P(r) function, Kratky plots, and Porod invariant. The highest quality scattering curve (5.0 mg ml$^{-1}$), as determined by the AutoRg and AutoGNOM functions in PRIMUS[64], was selected for ab initio modeling using scattering data up to q = 0.2 Å$^{-1}$. Since scattering curves are inherently ambiguous, we used the program Ambimeter[65] to assess whether ab initio modeling would yield reliable representations of the $CD22_{20-687}$ and $CD22_{20-687}$ + α-(2,6) sialyllactose structures in solution. We found that a single shape category (cylinder) was compatible with the scattering curves of $CD22_{20-687}$ and $CD22_{20-687}$ + α-(2,6) sialyllactose, resulting in ambiguity scores of 0.30 and 0, respectively, and deeming the ab initio reconstructions as potentially unique. Ten ab initio models were then independently generated using the program DAMMIF[32] without imposing symmetry. Ab initio models were aligned and averaged by using programs DAMAVER[66] and DAMFILT[66] to yield a final low-resolution model. Given the quality of the $x^2$-values (Supplementary Table 2) for all DAMMIF models and the fact that the scattering curves could only be modeled with one shape category, we concluded that the $CD22_{20-687}$ and $CD22_{20-687}$ + α-(2,6) sialyllactose samples have limited flexibility in solution and, therefore, the ab initio models are a good representation of their structures in solution.

**Isothermal titration calorimetry.** ITC experiments were performed using an Auto-ITC$_{200}$ (Malvern Instruments). ITC measurements of $CD22_{20-330}$ WT and mutants (K23S, R120A/E, R131A/E/K/Q) with α2-6 sialyllactose were collected using 70–110 µM of CD22 in the cell and 0.65–1.18 mM of α2-6 sialyllactose (Sigma-Aldrich) in the syringe. A total of 16 injections were performed with an injection volume of 2.5 µl, injection duration of 5, and 180 s spacing between injections. The cell temperature was set to 25 °C, with a stirring speed of 750 r.p.m. and a filter period of 5 s. All experiments were repeated at least in duplicates, and values were averaged and standard errors were calculated, based on a near 1:1 restricted binding stoichiometry. For Fab binding to CD22, 5 µM of CD22 was placed in the cell and 50 µM Fab was present in the syringe. A total of 16 injections were performed with an injection volume of 2.5 µl, injection duration of 5 s, and 180 s spacing between injections. The cell temperature was set to 35 °C, with a stirring speed of 750 r.p.m. and a filter period of 5 s. All experiments were repeated in triplicate, and values were averaged and standard errors were calculated.

**Biolayer interferometry.** The binding affinities of epratuzumab and pinatuzumab Fabs to CD22 were measured by BLI using the Octet RED96 BLI system (Pall ForteBio). Ni-NTA biosensors were hydrated in 1× kinetics buffer (1× PBS, pH 7.4, 0.002% Tween, 0.01% BSA) and loaded with 25 ng µl$^{-1}$ $CD22_{20-687}$ ($CD22_{20-687}$F, $CD22_{20-687}$S, $dCD22_{20-687}$, or $CD22_{20-687}$ glycan mutants) for 60 s at 1000 rpm. Biosensors were then transferred into wells containing 1× kinetics buffer to baseline for 60 s before being transferred into wells containing a serial dilution of Fab starting at 500 nM and decreasing to 62.5 nM. The 180 s association phase was subsequently followed by a 180 s dissociation step in 1× kinetics. Analysis was performed using the Octet software, with a 1:1 fit model. All experiments were repeated in triplicate, values were averaged, and standard errors were calculated.

**Data availability.** The crystal structures, EM reconstructions, and SAXS envelopes reported in this manuscript have been deposited in the Protein Data Bank, www.rcsb.org (PDB ID: 5VKJ, 5VKK, 5VKM, and 5VL3), the Electron Microscopy Data Bank, https://www.ebi.ac.uk/pdbe/emdb/ (EMDB ID: EMD-8704 and EMD-8705), and the Small Angle Scattering Biological Data Bank, www.sasbdb.org (SASBDB ID: SASDC76 and SASDC86), respectively. Other data are available from the corresponding author upon reasonable request.

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

## Acknowledgements

We thank X. Zuo and W. Shang for help with SAXS data collection and interpretation, A. Bosch for help with figure preparation, and J. Paulson, B. Treanor, and I. Wilson for discussions and critical comments on the manuscript. X-ray diffraction experiments described in this paper were performed using beamlines 08ID and 08BM at the Canadian

Light Source, which is supported by the Canada Foundation for Innovation, Natural Sciences and Engineering Research Council of Canada, the University of Saskatchewan, the Government of Saskatchewan, Western Economic Diversification Canada, the National Research Council Canada, and the Canadian Institutes of Health Research. SAXS experiments were performed at beamlines 12-ID and 18-ID of the Advanced Photon Source, a U.S. Department of Energy (DOE) Office of Science User Facility operated for the DOE Office of Science by Argonne National Laboratory under Contract No. DE-AC02-06CH11357. We would like to acknowledge the Structural & Biophysical Core Facility, The Hospital for Sick Children, for access to the ITC and Octet RED96 BLI instruments. J.E.-O. was supported by Banting Postdoctoral Fellowship BPF-144483 from the Canadian Institutes of Health Research. T.S. is a recipient of a Canada Graduate Scholarship Master's Award from the Canadian Institutes of Health Research. This work was supported by operating grant PJT-148811 (J.-P.J.) from the Canadian Institutes of Health Research.

## Author contributions

The following outlines author contributions. Experimental conception and design: J.E.-O., T.S., J.-P.J.; data acquisition: J.E.-O., T.S., H.C.; analysis of data: J.E.-O., T.S., M.T.M.-J., S.B., A.G., J.L.R., J.-P.J.; drafting the article or revising it critically for important intellectual content: J.E.-O., T.S., J.-P.J.

## Additional information

**Competing interests:** The authors declare no competing financial interests.

