## [Peer Review File · Nature Communications]

Reviewers' Comments:

Reviewer #1:

Remarks to the Author:

Ereno-Orbea and co-authors report a detailed structural account of CD22, which includes crystallographic studies on a CD22 fragment, its co-complex with a sialic acid ligand, a solution-based and EM structure of its entire ectodomain, and a co-complex with a Fab fragment derived from a therapeutic antibody. Judging from the protein chemistry/mutagenesis required, it would appear that CD22 was a difficult target to extract structural information from, and the authors are to be congratulated in putting together a comprehensive portrait of CD22 structure. The structure determination is technically well done and the figures complement the structural description well. The structures provide a platform for a nice comparison with the other siglec family members, of which there is a relative abundance of information. Moreover, speculation into how the structural features of CD22 leads to biological function is presented. Examining Figure 4, and associated sup. data, the concept of "semi-rigid" appears rather subjective – are there any objective criterion from which this "rigid feature" can be ascertained? Overall, I feel that the work is appropriate for publication in Nature Comms.

Minor points

- 1) BSA values reported are too accurate and supp tables need to be tempered here with reporting such values, especially given the low resolution of the Fab-CD22 complex
- 2) Structures are refined well, but can the authors clarify what "ligand" refers to- especially in the Fab co-complex.

Reviewer #2:

Remarks to the Author:

this is a great manuscript and was a pleasure to read. The topic is interesting.

Data is clearly presented and accompanied by relevant and thoroughly described supplementary information. The description of SAXS data collection and interpretation is exemplary. The authors have followed the general publication guidelines especially in terms of addressing sample purity and adding the SAXS data table.

I highly recommend this manuscript for publication.

very minor comments

*Page 6

Citation format issue: Bakker et al., 2002

*page 33,

* Citation format issue:: PRIMUS (71);

* Citation format issue: for Ambimeter (Petoukhov & Svergun)

*please check if ambiguity score is really zero

*Deposition

In addition to the x-tal and EM data, the SAXS data can also be deposited at Sasbdb (<http://sasbdb.org/>) or BioIsis (<http://www.bioisis.net/>)

Reviewer #3:

Remarks to the Author:

The manuscript by Ereno-Orbea et al., describes a number of structures of CD22 in complex with both bound ligands and a therapeutic antibody. There are a number of studies here which combined significantly advances our understanding of this important system. Therefore

fundamentally I think this study would be of interest to your readers. The combination of techniques such as X-ray crystallography, SAXS and EM provides a powerful approach to understanding this system. There are however a number of points which need to be addressed which I have detailed below. The main emphasis is that as written the paper can be confusing and at times more data needs to be shown, for example the electron density for the inhibitors is only shown for the inhibitor but surrounding residues should also be shown. Areas of the manuscript which I feel could be improved are (ordered broadly in when they appear in the manuscript);

1) The introduction would benefit from the appropriate sub-heading.

2) The introduction is very brief at only 38 lines of which 11 lines are simply stating the results from the paper and outlining its objective. If this paper is to appeal to a wide audience it will benefit from a more extended introduction. For example in the results the authors talk about "d1 adopting a V-type fold", what is d1? There is an extensive discussion in the paper but without sufficient introduction this loses its impact and context.

3) The authors talk about buried surface area for Siglec-4 and Siglec-5, where do these numbers come from, should there be a reference, which structures are being used for the comparison? If they calculated themselves what did they use?

4) The authors state how N101 is "buried in a hydrophobic environment...which helps explain why it is essential for CD22 expression". Do the related Siglec family which have a glycan at a similar position (5 & 7) also have this glycan within the solved crystal structures?

5) The top right panel shows the density for the bound ligand, however there is no density for the surrounding residues. I understand how for clarity it can be beneficial to just show the ligand density and that the use of an omit map is a good way to show a lack of bias. However, it is important that this is shown in the context of the rest of the map, is this a strong feature, how does it compare to the neighbouring residues. The figure should be remade to show this, or one should be placed in Supp. Materials to show this.

6) The authors use crystal soaking to bind α 2-6 sialyllactose, it is shown in Figure 3 bound, but why is the electron density not shown? It is important that the density map is shown, especially when discussing areas of strong and weak density. How ambiguous is the fitting of α 2-6 sialyllactose, especially when discussing the interaction with Y64. This figure should be included possibly in supp. Figures. Also are there any interactions with crystal contacts that may be influencing binding?

7) It might be beneficial to state what the mutations in Y64 are, rather than having to read the reference just to see, was it simply Ala scanning, a closely related Phe?

8) The authors note how highly similar the bound and unbound CD22 structures are with an r.m.s.d. of 0.35, is this surprising given it's a crystal soaking experiment and therefore the full conformational changes may not be possible due to lattice constraints. Does the loop in Siglecs 4 and 7 undergo a conformational change because its not constrained by the lattice environment? There are many examples in the literature where crystal soaking does not reflect the full dynamics of a system. Maybe this caveat should be discussed?

9) Supp Figure 4. When fitting the components in the negative stain EM envelope it looks like they could be fitted the other way round ie with d1-d3 at the base, does this other orientation fit? Also what do the numbers at the top right of the classes represent in panel (e)?

10) Page 9, line 13, This should be Supplementary Fig. 6e.

11) When discussing the structure of the CD22-Epratuzumab crystal complex there is no figure for the electron density map to show the quality of the data, this would be a welcome addition.

12) In Figure 6, what is the red dashed line and blue dashed arrow representing?

Reviewer #4:

None

RESPONSE TO REVIEWERS' COMMENTS:

Reviewer #1 (Remarks to the Author):

Ereno-Orbea and co-authors report a detailed structural account of CD22, which includes crystallographic studies on a CD22 fragment, its co-complex with a sialic acid ligand, a solution-based and EM structure of its entire ectodomain, and a co-complex with a Fab fragment derived from a therapeutic antibody. Judging from the protein chemistry/mutagenesis required, it would appear that CD22 was a difficult target to extract structural information from, and the authors are to be congratulated in putting together a comprehensive portrait of CD22 structure. The structure determination is technically well done and the figures complement the structural description well. The structures provide a platform for a nice comparison with the other siglec family members, of which there is a relative abundance of information. Moreover, speculation into how the structural features of CD22 leads to biological function is presented.

We thank the reviewer for these encouraging comments.

Examining Figure 4, and associated sup. data, the concept of “semi-rigid” appears rather subjective – are there any objective criterion from which this “rigid feature” can be ascertained?

On p.9, we present EM and SAXS data that indicate a small range of conformations adopted by the CD22 ectodomain. From this data, we conclude that the CD22 ectodomain adopts an elongated tilted structure with low flexibility. We agree with the reviewer that “semi-rigid” is subjective, consequently we removed this nomenclature. Instead, we now highlight the low flexibility observed for the CD22 ectodomain, and put this finding in contrast to other multi-Ig domain molecules with *cis/trans* binding modes that show significant flexibility e.g. protein tyrosine phosphatase sigma (RTP σ) involved in synaptogenesis (p.9).

Overall, I feel that the work is appropriate for publication in Nature Comms.

Minor points

1) BSA values reported are too accurate and supp tables need to be tempered here with reporting such values, especially given the low resolution of the Fab-CD22 complex

We have now removed the decimal from the BSA values reported in Supplementary Tables 1 and 3.

2) Structures are refined well, but can the authors clarify what “ligand” refers to- especially in the Fab co-complex.

We thank the reviewer for pointing out the incorrect use of “ligand” in Table 1. We now use the correct term “hetero” atoms.

Reviewer #2 (Remarks to the Author):

this is a great manuscript and was a pleasure to read. The topic is interesting. Data is clearly presented and accompanied by relevant and thoroughly described supplementary information. The description of SAXS data collection and interpretation is exemplary. The authors have followed the general publication guidelines especially in terms of addressing sample purity and adding the SAXS data table.

I highly recommend this manuscript for publication.

We are grateful to the reviewer for his/her support for our manuscript.

very minor comments

*Page 6

Citation format issue: Bakker et al., 2002

*page 33,

* Citation format issue:: PRIMUS (71);

* Citation format issue: for Ambimeter (Petoukhov & Svergun)

These citation format issues have now been corrected.

*please check if ambiguity score is really zero

Independent of the version of the software that is used to perform the analysis (versions 2.8.1 or 2.7), the ammeter score is zero for CD22₂₀₋₆₈₇ in molar excess of α -2-6 sialyllactose, and of low ambiguity (0.30) for CD22₂₀₋₆₈₇.

*Deposition

In addition to the x-tal and EM data, the SAXS data can also be deposited at Sasbdb (<http://sasbdb.org/>) or BioIsis (<http://www.bioisis.net/>)

We have now deposited the SAXS data at Sasbdb (<http://sasbdb.org/>). We provide the accession numbers for these dataset in the Data availability section of the manuscript (p.23-24).

Reviewer #3 (Remarks to the Author):

The manuscript by Ereno-Orbea et al., describes a number of structures of CD22 in complex with both bound ligands and a therapeutic antibody. There are a number of studies here which combined significantly advances our understanding of this important system. Therefore fundamentally I think this study would be of interest to your readers. The combination of techniques such as X-ray crystallography, SAXS and EM provides a powerful approach to understanding this system. There are however a number of points which need to be addressed which I have detailed below. The main emphasis is that as written the paper can be confusing and at times more data needs to be shown, for example the electron density for the inhibitors is

only shown for the inhibitor but surrounding residues should also be shown. Areas of the manuscript which I feel could be improved are (ordered broadly in when they appear in the manuscript);

1) The introduction would benefit from the appropriate sub-heading.

The Introduction sub-heading has been added.

2) The introduction is very brief at only 38 lines of which 11 lines are simply stating the results from the paper and outlining its objective. If this paper is to appeal to a wide audience it will benefit from a more extended introduction. For example in the results the authors talk about “d1 adopting a V-type fold”, what is d1? There is an extensive discussion in the paper but without sufficient introduction this loses its impact and context.

We thank the reviewer for this suggestion. To increase the appeal of our manuscript to a wide audience, we now more thoroughly introduce CD22 sequence features and the biological roles of sialic acid glycans in health and disease (p.3-4). We added the following text: “CD22 is a single-spanning membrane glycoprotein of 140 kDa on the surface of B cells. The extracellular domain (ECD) of CD22 is comprised of seven immunoglobulin (Ig) domains (d1-d7) and 12 putative N-linked glycosylation sites. The most N-terminal domain (d1) is of predicted V-type immunoglobulin (Ig)-like fold and recognizes sialic acids containing α 2,6-linkages⁵. While human CD22 binds preferentially to Sia N-acetyl neuraminic acid (Neu5Ac), murine CD22 has higher specificity towards the non-human N-glycolyl neuraminic acid (Neu5Gc)⁶, highlighting species-dependent specificities for CD22 ligand recognition. Moreover, cell-surface sialylated glycans can be modified, typically at the 4, 6, 7 or 9 hydroxyl positions which can alter their binding specificities to CD22^{7,8}. Some of these changes are associated with cellular dysregulation. As examples, O-acetylation at the 9-hydroxyl position has been implicated in autoimmunity^{7,8} and in progression of childhood acute lymphoblastic leukemia (ALL)⁹.”

3) The authors talk about buried surface area for Siglec-4 and Siglec-5, where do these numbers come from, should there be a reference, which structures are being used for the comparison? If they calculated themselves what did they use?

We have added a reference to the manuscripts and the PDB accession code reporting these structures in the text (p.6): “In the crystal structures of Siglecs -4 (PDB ID: 5LFR) and -7 (PDB ID: 1O7S), the equivalent N101 glycan also buries a significant surface area on the protein (589 Å² and 317 Å², respectively)^{23,25,26}.” In the Materials and Methods section, we now describe how buried surface area calculations were performed (p.20): “All buried surface area values reported were calculated using EMBL-EBI PDBePISA²⁵.”

4) The authors state how N101 is “buried in a hydrophobic environment... which helps explain why it is essential for CD22 expression”. Do the related Siglec family which have a glycan at a similar position (5 & 7) also have this glycan within the solved crystal structures?

As shown in Supplementary Figure 2a, the crystal structure of mammalian cell-expressed Siglec-4 has a glycan at the corresponding position (Pronker *et al.*, Nat Commun 2016). CHO Lec1 mammalian cells were used for expression of Siglec-7, and the corresponding glycan was well ordered in the electron density (Alphey *et al.*, J Biol Chem 2003). For crystal structure determination, Siglec-5 was expressed in *E.coli* and therefore was not glycosylated. However, it required systematic refolding from inclusion bodies to obtain a well-folded protein, perhaps hinting at a role of glycosylation for expression of soluble protein (Zhuravleva *et al.*, J Mol Biol 2008). We have now added a sentence on p.6 to highlight the presence of a well-ordered glycan at this position in related Siglecs expressed in mammalian cells to further strengthen our argument of its role in expression: “In the crystal structures of Siglecs -4 (PDB ID: 5LFR) and -7 (PDB ID: 1O7S), the equivalent N101 glycan also buries a significant surface area on the protein (589 Å² and 317 Å², respectively)^{23,25,26}.”

5) The top right panel shows the density for the bound ligand, however there is no density for the surrounding residues. I understand how for clarity it can be beneficial to just show the ligand density and that the use of an omit map is a good way to show a lack of bias. However, it is important that this is shown in the context of the rest of the map, is this a strong feature, how does it compare to the neighbouring residues. The figure should be remade to show this, or one should be placed in Supp. Materials to show this.

At the reviewer’s request, we have now added a composite omit map of the N101 glycan and surrounding residues in Supplementary Figure 8a to depict the quality of the electron density map in this region.

6) The authors use crystal soaking to bind α2-6 sialyllactose, it is shown in Figure 3 bound, but why is the electron density not shown? It is important that the density map is shown, especially when discussing areas of strong and weak density. How ambiguous is the fitting of α2-6 sialyllactose, especially when discussing the interaction with Y64. This figure should be included possibly in supp. Figures. Also are there any interactions with crystal contacts that may be influencing binding?

We thank the reviewer for raising this point. Fitting of the α2-6 sialyllactose was unambiguous and there are minimal crystal contacts in this region that may influence binding. At the reviewer’s request, we have now added a composite omit map of the ligand and binding-site residues in Supplementary Figure 8b to depict the quality of the electron density map in this region, and the strong electron density for Neu5Aca(2-6)Gal.

7) It might be beneficial to state what the mutations in Y64 are, rather than having to read the reference just to see, was it simply Ala scanning, a closely related Phe?

We have now added more details on the mutagenesis performed in van der Merwe *et al.*, J Biol Chem 1996 that corroborates our structural delineation of the CD22 ligand binding site (p.7): “E126 and W128 make key contacts with Sia (Figures 3a-b), and these interactions corroborate previous biochemical studies that delineated the CD22 binding site by mutagenesis of these residues to lysine and arginine, respectively.” We now also note in

this section that the equivalent position of human CD22 Y64 in mouse is F68, making the aromatic ring a conserved feature across CD22 in different species to participate in stacking interactions with the hydrophobic face of galactose of α 2-6 Sia ligands.

8) The authors note how highly similar the bound and unbound CD22 structures are with an r.m.s.d. of 0.35, is this surprising given it's a crystal soaking experiment and therefore the full conformational changes may not be possible due to lattice constraints. Does the loop in Siglecs 4 and 7 undergo a conformational change because its not constrained by the lattice environment? There are many examples in the literature where crystal soaking does not reflect the full dynamics of a system. Maybe this caveat should be discussed?

We agree with the reviewer that crystal contacts can significantly constrain the dynamics of a system during ligand soaking experiments. The CD22 binding site is largely free of crystal contacts, which allowed to successfully soak in the α 2-6 sialyllactose ligand. Nonetheless, surrounding loops do participate in forming the crystal lattice. In Siglecs 4 and 7, C-C' loop is disordered in the absence of the ligand and it adopts an ordered conformation in its presence, as illustrated in Supplementary Figure 2b. Although in CD22 the C-C' loop (i.e. C1/C2 β -hairpin) is involved in some interactions that facilitate crystal packing, we propose that the extensive intra-molecular forces of the C1/C2 β -hairpin primarily drive the pre-disposed nature of the CD22 ligand binding site. We now illustrate intra- and inter-molecular interactions in the context of crystal packing in Supplementary Figure 2c and emphasize the minimal role for crystal packing in restricting full motion of the CD22 ligand binding site on p.8: "Extensive intra-molecular H-bonds between C1 and C2 in the β -hairpin and van der Waals interactions between F71 and M129 are major determinants of the preformed binding site (Supplementary Fig. 2c). We note minimal interactions within the crystal lattice that might have artifactually constrained the C1/C2 β -hairpin in our soaking experiments." Similarly, Zhuravleva *et al.*, J Mol Biol 2008 proposed that the "closed" conformation of the C-C' loop observed in Siglec-5 in the absence or presence of the ligand is partly attributed to intra-molecular interactions between C-C' and C'-D loop.

9) Supp Figure 4. When fitting the components in the negative stain EM envelope it looks like they could be fitted the other way round ie with d1-d3 at the base, does this other orientation fit? Also what do the numbers at the top right of the classes represent in panel (e)?

The reviewer is correct that the CD22 d1-d3 crystal structure has alternate fittings in the EM reconstruction. However, two main lines of evidence suggest our proposed model is more likely than its alternatives. First, as we now state in the Materials and Methods section on p.21: "Fitting of the CD22 d1-d3 crystal structure in the CD22₂₀₋₆₈₇ EM volume using UCSF Chimera ⁶⁷ favored one orientation (higher fitting score of 0.905 with 0 atoms outside of map contour) compared to when it was fitted in an alternative orientation at the base (0.890 fitting score with 27 atoms outside of map contour)." This is largely because the top of the EM reconstruction as represented in Supplementary Figure 4 is larger, which helps accommodate the bigger V-type Ig domain fold (d1) compared to smaller C1- and C2-type Ig domains that correspond to CD22 d2-d7. Second, this larger volume at the top of CD22 is also near additional density observed for epratuzumab Fab in the CD22-

epratuzumab Fab EM reconstruction which we know from our co-crystal structure binds at the interface between CD22 d2-d3 (Fig. 5a). On p.21, we now also clarify that: “Fitting of the CD22_{20-330,4Q}-epratuzumab Fab crystal structure in the CD22₂₀₋₆₈₇-epratuzumab Fab EM volume using UCSF Chimera⁶⁷ was unambiguous.” The numbers in Supplementary Figure 4e represent the number of particle images in each class. The number has been moved to the bottom right of the classes for consistency with Fig. 4a, and a description has been added to the figure legend.

10) Page 9, line 13, This should be Supplementary Fig. 6e.

We have updated the call-out of this Supplementary Figure.

11) When discussing the structure of the CD22-Epratuzumab crystal complex there is no figure for the electron density map to show the quality of the data, this would be a welcome addition.

We have now added a composite omit map of the CD22-epratuzumab antigen-antibody interface in Supplementary Figure 8c to depict the quality of the electron density map of this co-crystal structure.

12) In Figure 6, what is the red dashed line and blue dashed arrow representing?

A description has been added to the figure legend detailing how the red dashed line represents inhibition of BCR signaling, which is shown as a blue dashed arrow.